# MASKED AUTODECODER IS EFFECTIVE MULTI-TASK VISION GENERALIST

## ABSTRACT

Inspired by the success of general-purpose models in NLP, recent studies attempt to unify different vision tasks in the same sequence format and employ autoregressive Transformers for sequence prediction. They apply uni-directional attention to capture sequential dependencies and generate task sequences recursively. However, such autoregressive Transformers may not fit vision tasks well, as vision task sequences usually lack the sequential dependencies typically observed in natural languages. In this work, we design Masked AutoDecoder (MAD), an effective multi-task vision generalist. MAD consists of two core designs. First, we develop a parallel decoding framework that introduces bi-directional attention to capture contextual dependencies comprehensively and decode vision task sequences in parallel. Second, we design a masked sequence modeling approach that learns rich task contexts by masking and reconstructing task sequences. In this way, MAD handles all the tasks by a single network branch and a simple cross-entropy loss with minimal task-specific designs. Extensive experiments demonstrate the great potential of MAD as a new paradigm for unifying various vision tasks. MAD achieves superior performance and inference efficiency compared to autoregressive counterparts while obtaining competitive accuracy with task-specific models.

## 1    INTRODUCTION

Computer vision covers various concepts, such as localization, classification, and description, leading to a wide variety of highly structured outputs in different vision tasks, i.e., object detection, instance segmentation, keypoint detection, image captioning, etc. Following natural language processing (NLP), recent methods (Lu et al., 2022; Chen et al., 2022a; Wang et al., 2023; 2022; Kolesnikov et al., 2022) attempt to unify different vision tasks in an autoregressive sequence-to-sequence framework as illustrated in the upper part of Fig. 1. They first model different vision tasks in the same sequence format, such as a sequence of coordinate and class label tokens for object detection, a sequence of contour coordinate tokens for image segmentation, or a sequence of descriptive sentences for image captioning. Additionally, the autoregressive Transformers (Brown et al., 2020; Radford et al., 2018; 2019), with its specially designed uni-directional attention to capture sequential dependencies, are employed to recursively predict these vision task sequences.

Despite the success, the autoregressive approach often struggles on vision tasks due to two major factors: (1) *The discrepancy between vision and language*. Language task sequences (Brown et al., 2020; Touvron et al., 2023) heavily follow sequential dependencies while vision task sequences may not, e.g., the next word prediction in a sentence highly depends on its preceding texts, while the pixel prediction in segmentation tasks largely depends on its neighboring content instead of merely previous ones. The autoregressive approach, with uni-directional attention, can well capture sequential dependencies for language tasks but may not fit well with vision tasks. (2) *Computation Efficiency*. The autoregressive approach predicts tokens in a sequence recursively, which is computation-intensive. The two factors might limit the model performance and efficiency, hindering the application of the autoregressive approach to vision tasks.

One possible solution for mitigating the above two issues is to explore bi-directional attention and parallel prediction for sequence modeling. This design leads to a customized Transformer that is capable of capturing more comprehensive dependencies and decodes the task sequence from scratch in parallel. However, such decoding process from scratch may struggle while modeling task contexts

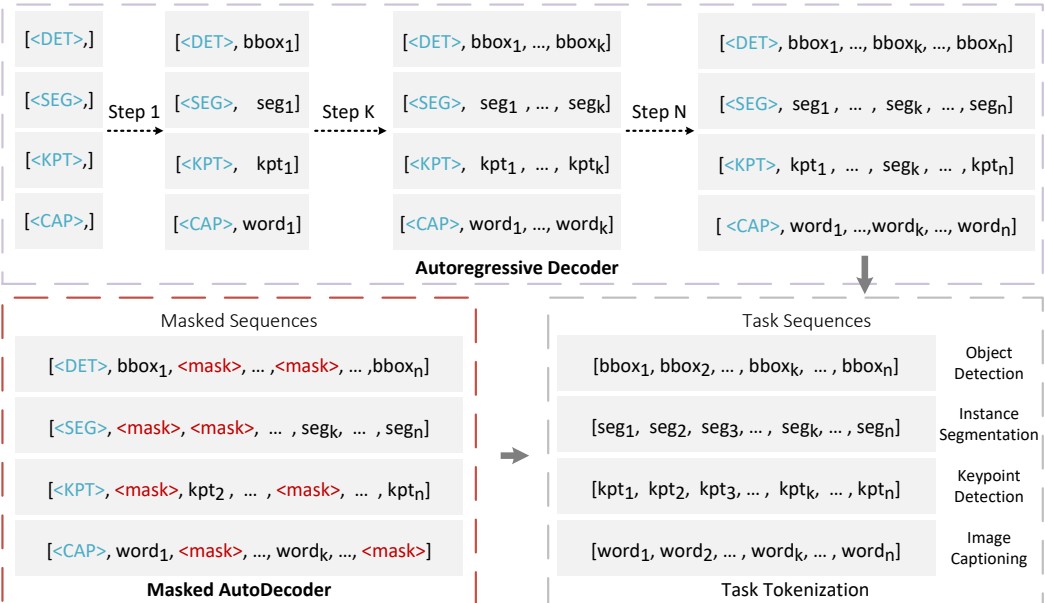

Figure 1: Unified Sequence-to-sequence Modeling of Vision Tasks. The traditional *Autoregressive Decoder* adopts a multi-step recursive process for prediction. It generates task sequences token by token and utilizes uni-directional attention where each token can only attend to its previous ones. The generation step is repeated $N$ times depending on the length of sequences. Our proposed *Masked AutoDecoder* (MAD) adopts parallel decoding where each token can perceive all other tokens in the sequence. In addition, by masking and reconstructing task sequences, MAD is capable of capturing rich task contexts for multi-task learning. Tokens in blue indicate task prompts, and *Task Sequences* are simplified with details to be described in the ensuing sections.

as sequences from different tasks highly vary in patterns, lengths, token vocabularies, etc., which will impede network convergence and result in inferior performance for multi-task learning.

Driven by the above analysis, we present Masked Decoder (MAD), an effective sequence-based generalist for vision tasks. As illustrated in the bottom-left part of Fig. 1, MAD masks tokens randomly from the task sequences and reconstructs the masked tokens based on the unmasked ones and image features, which provides rich task contexts for modeling disparate task sequences. In addition, it adopts an encoder-decoder Transformer architecture with bi-directional attention that leverages comprehensive dependencies in vision tasks to effectively decode task sequences in parallel. These designs enable a more efficient and effective multi-task learning framework that performs multiple vision tasks in a single architecture. Our experiments with four tasks (object detection, instance segmentation. key-point detection, and image captioning) on the COCO dataset demonstrate that a simple MAD can achieve competitive accuracy and efficiency compared to both task-customized approaches and existing generalist models.

## 2 RELATED WORKS

**Vision Generalist Models** Learning a vision generalist model capable of handling multiple vision tasks using a shared architecture has long been a goal in computer vision. Inspired by the success of unified sequence-to-sequence based transformer framework (Devlin et al., 2018; Radford et al., 2018; 2019) in natural language processing (NLP), recent works (Alayrac et al., 2022; Wang et al., 2022; Reed et al., 2022; Chen et al., 2022b) extend this framework to the field of computer vision and model various vision tasks in a unified sequence-to-sequence autoregressive paradigm. The pioneering works (Cho et al., 2021; Li et al., 2022; Zhu et al., 2022) mainly focus on high-level semantic tasks, such as image captioning, visual question answering, image-text matching, and etc., considering their intrinsic correlation with language. In pursuit of unifying more vision tasks, especially for those involving dense predictions, Pix2seq (Chen et al., 2021; 2022a) and UniTab (Yang et al., 2022)

discrete object positions as a series of coordinate tokens to enable the localization capability for generalist models. Unified-IO (Lu et al., 2022) and UViM (Kolesnikov et al., 2022) encode the per-pixel targets into semantic tokens for vision tasks that require outputs as images, such as depth estimation or panoptic segmentation. Uni-PercieverV2 (Li et al., 2023) equips an additional region proposal network to generate sequence predictions for object detection and instance segmentation. VisionLLM (Wang et al., 2023) leverages LLM to enable flexible task output formats. Different from these methods which focus on customizing and extending more vision tasks in a sequence-based autoregressive framework, we demonstrate that such a framework may not fit well for vision tasks. Our masked Auto-decoding pursues a conceptually different direction and learns diverse task contexts in parallel via masked sequence modeling, leading to a more efficient and effective vision generalist.

**Masked Signal Modeling** The paradigm of learning rich representations via masking and reconstructing has been widely explored in both fields of NLP and computer vision. In NLP, through masking and recovering language sentences, models like BERT (Devlin et al., 2018) and its variants (Liu et al., 2019; Lan et al., 2019) successfully pre-train models capable of generalizing to a broad range of NLP tasks. In computer vision, such a paradigm also leads to multiple masked image modeling (MIM) (Gao et al., 2022; Dong et al., 2022) and masked video modeling (MVM) techniques. For example, BEIT (Bao et al., 2021) explores MIM by recovering the masked image into visual tokens from discrete VAE (Ramesh et al., 2021). SimMIM (Xie et al., 2022), MaskFeat (Wei et al., 2022), and MAE (He et al., 2022) incorporate low-level visual signals, such as RGB pixel value or the feature descriptor HOG (Dalal & Triggs, 2005), as the reconstruction targets. VideoMAE (Feichtenhofer et al., 2022) encodes the corrupted video and learns to recover both spatial and temporal signals. The above methods employ masked signal modeling as a self-supervised task, aiming to learn to auto-encode rich representations for downstream tasks. Different from them, we propose masked auto-decoder (MAD), exploring masked sequence modeling for decoding task sequences from its masked variants. Our approach is close to non-autoregressive translation (Ghazvininejad et al., 2019; Gu et al., 2017) in NLP, but it has very different intrinsic objectives - non-autoregressive translation exploits parallel decoding to improve translation efficiency, while MAD aims to model diverse task contexts for learning multi-task vision generalists.

## 3 METHODS

Our proposed unified generalist framework consists of three key components: (1) Unified tokenization of diverse input and output sequences for different tasks; (2) Masked auto-decoding framework for modeling task contexts; (3) An architecture that decodes desired task sequences based on image features. We introduce these components in the following sections.

### 3.1 TASK TOKENIZATION

In this work, we consider four vision-related tasks, including object detection, instance segmentation, keypoint detection, and image captioning. These tasks require the model's ability from classification to localization, from vision to language, and from image-level to pixel-level recognition. Therefore, A comprehensive vocabulary is essential for dealing with such sophisticated problems. Our vocabulary comprises five parts, including prompt tokens to distinguish tasks, coordinate tokens for localization, category tokens for classification, task-related special tokens, and word tokens for captioning, more detail to be elaborated in the ensuing subsections.

For object detection, following Pix2Seq (Chen et al., 2021), we convert bounding boxes into a sequence of tokens consisting of discrete coordinates and categories by the order of $[x_{min}, y_{min}, x_{max}, y_{max}, class]$. As described in Fig. 2, we construct a sequence consisting of $N$ noise objects, and then randomly replace and inject the ground truth object in the sequence. The $< Detection >$ prompt tokens are added before the sequence to identify the task. We set $N$ at 100 by default.

For instance segmentation, we directly predict the pixel mask following Mask R-CNN (He et al., 2017). The bit masks of the size $M \times M$ are flattened and transformed into sequences consisting of $< Foreground >$ tokens and $< Background >$ tokens. We concatenate the prompt sequence consisting of task token $< Segmentation >$, bounding box coordinate tokens, and a class token to identify different instances.

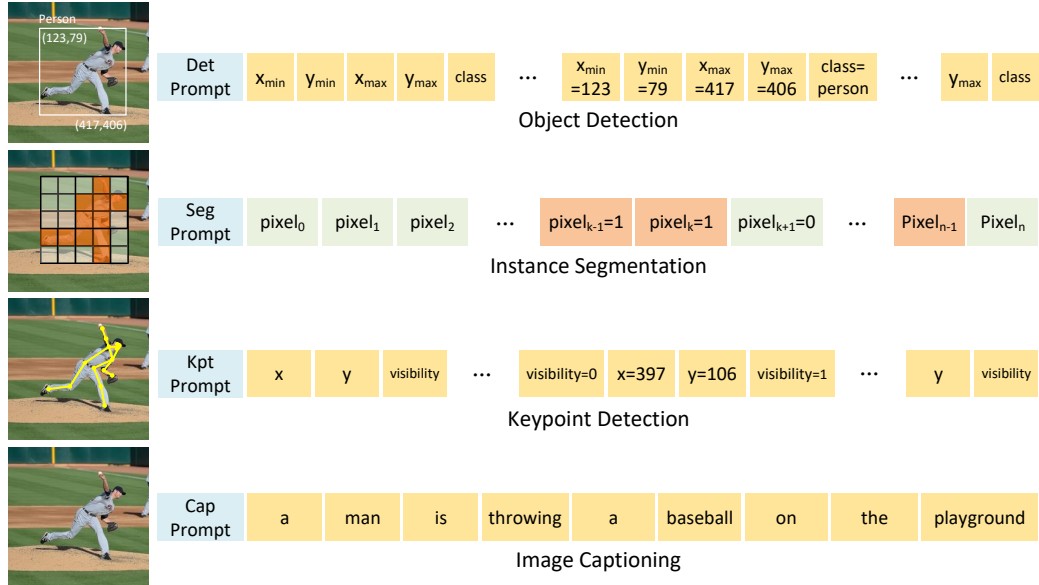

Figure 2: Four vision tasks are tokenized into a unified sequence format. Sequences for object detection consist of coordinate tokens and class tokens. For instance segmentation, we adopt two customized tokens to represent foreground pixels (pixel=1) and background pixels (pixel=0). Keypoint detection shares the same coordinate tokens to object detection, with two additional tokens for visible and invisible keypoints. We adopt sentence-piece model (Kudo & Richardson, 2018) to tokenize captioning sentences into subword token sequences, but show words for simplicity.

For keypoint detection, we predict the coordinates and visibility for each keypoint of the person instance. It can thus be represented as a sequence of $[x, y, visibility, x, y, visibility, ...]$. We adopt two tokens $< Visible >$ and $< Invisible >$ to depict the visibility. The keypoints are arranged by the default order as in COCO dataset (Lin et al., 2014). For the occluded keypoints, we replace their coordinate tokens with random coordinates within the bounding box. We utilize the sequence $[< Keypoint >, x_{min}, y_{min}, x_{max}, y_{max}, person]$ to prompt keypoint detection task, where the coordinates in the prompt indicate the bounding box of the corresponding person.

For captioning, we adopt a pre-trained sentence-piece model (SPM) (Kudo & Richardson, 2018) to convert a caption into a sequence of discrete tokens. We randomly replace one of the tokens in the transferred sequence with a random word token for sequence augmentation. All the sequences are padded or truncated to a length of 20 tokens. The $< Caption >$ token is adopted as the prompt.

## 3.2 MASKED AUTODECODING

**Masked Training** We propose Masked Auto-Decoding for multi-task sequence modeling. We randomly sample a subset of target tokens and mask the remaining ones. The sampling follows a uniform distribution. The masked tokens are replaced by special $< Mask >$ tokens, which are shared among all tasks. During training, we adopt two kinds of sequences for each task, a fully masked sequence and a partly masked sequence.

The reconstruction of fully masked sequences establishes a basis to train a unified decoder, which is, learning to decode multi-task sequences with only the prompts. Therefore, all the tokens, except those in the prompt sequences, are replaced with $< Mask >$ before being fed into the decoder. The training objective is to reconstruct the desired task sequences based on task prompts. However, different from the autoregressive approach where each task sequence is specified by its corresponding input sequence, a fully masked sequence in auto-decoding might match multiple similar task sequences, such as differently arranged objects for object detection or similar captioning sentences per image for image captioning. Randomly choosing the reconstruction target each time might hinder convergence. Hence, we adopt *Hungarian Matching* (Kuhn, 1955) to construct the task sequences for object

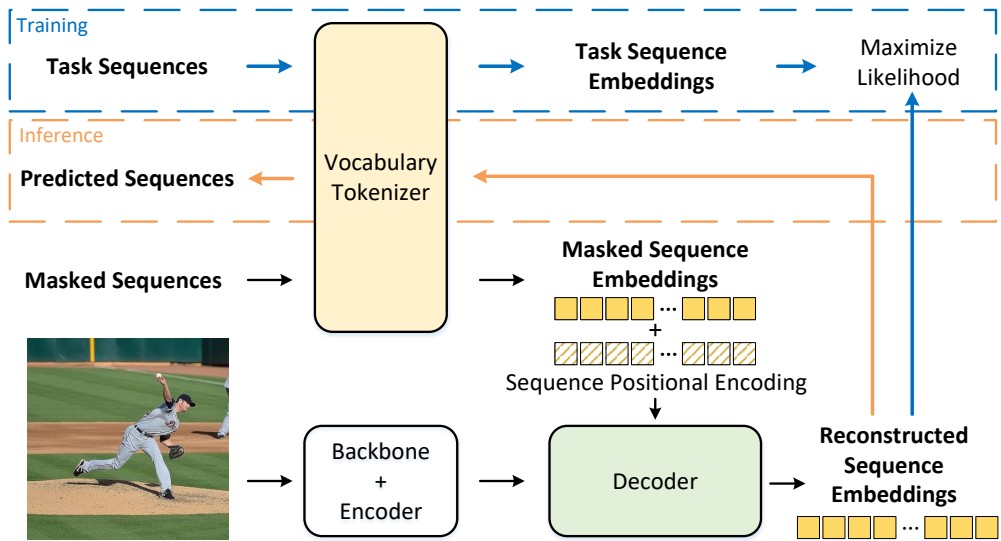

Figure 3: The proposed MAD architecture. MAD consists of three major parts, a *Backbone + Encoder* to extract the representation of the input images, a *Decoder* that processes masked sequences for prediction, and a *Vocabulary Tokenizer* that transfers between token sequences and embeddings. We randomly mask *Task Sequences* for *Masked Sequences* during training, and take fully *Masked Sequences* as input for inference.

detection and image captioning. For instance segmentation and keypoint detection, the original unmasked sequences are adopted as targets since their prompt sequences with object locations and categories are able to specify the unambiguous task sequences.

However, when all the tokens are masked, it is difficult for the model to distinguish different task sequences based on only a few prompt tokens. We thus leverage partly masked sequences to alleviate this issue. The unmasked tokens provide rich cues for the pattern of different task sequences, which help the decoder capture diverse task contexts. During training, both fully and partly masked sequences are concatenated together and decoded in parallel.

The MAD task is greatly inspired by the self-supervised masked auto-encoding approach in both language and vision domains, which learns and encodes informative representation by reconstructing masked content. We expand this idea of masked modeling to decode multi-task sequences in computer vision. This simple method, by modeling corrected sequences and predicting missing tokens, enables MAD to learn distinct task contexts and inter-sequence dependencies for vision tasks.

**Masked Inference** During inference, we conduct multi-stage masked decoding to refine the prediction. With the initial prediction recovered from the fully masked sequence, we sample part of the sequences and replace them with mask tokens again. The corrupting sequences are then fed to the decoder for reconstruction. We directly ensemble predictions from masked tokens to their original tokens to obtain more accurate predictions.

## 3.3 ARCHITECTURE

Our goal is to build a single model that is capable of handling different vision-related tasks within a unified sequence paradigm with little task-customized designs. Hence, we adopt a simple encoder-decoder transformer architecture, which has been proven successful in handling sequences with variable lengths in both natural language processing and computer vision tasks. As shown in Fig. 3, the overall architecture of MAD consists of three main components, a backbone network with the encoder to extract image features, a decoder to reconstruct the masked sequences, and a vocabulary tokenizer that transforms between token sequences and embedding.

**Backbone and Transformer Encoder.** Given an input image $I \in \mathbb{R}^{H \times W \times 3}$, a backbone network is adopted to generate a low-resolution image feature with a stride of 32. The encoder then takes the image feature, adds 2D positional encodings, and processes the feature via a series of encoder layers consisting of a self-attention module and feed-forward network (FFN). The image feature is then injected into the decoder as a condition to decode task sequences.

**Vocabulary Tokenizer.** We leverage vocabulary tokenizer to transform between token sequences and sequence embeddings. It maintains a vocabulary of embeddings with a dimension of $D$, which corresponds to all tokens as described in Sec. 3.1. Before being fed into the decoder, the discrete *Masked Sequences* of a length $L$ are converted into *Masked Sequence Embeddings* $E \in \mathbb{R}^{L \times D}$ by directly indexing the vocabulary. After being recovered by the decoder, we adopt cosine similarity to transform the *Reconstructed Sequence Embeddings* back to the *Predicted Sequences*.

**Transformer Decoder.** The decoder follows standard architecture, reconstructing *Masked Sequence Embeddings* through self-attention, cross-attention, and FFN layers. To address the sequence order, we introduce learned *Sequence Positional Encoding* and add them to the input embeddings before each attention layer in the decoder. The *Sequence Positional Encodings* are shared among all the tasks and are truncated according to the length of different task sequences. Unlike existing autoregressive methods (Chen et al., 2022a; Lu et al., 2022) that adopt uni-directional masks in self-attention layers and generate only one token at a time, our model decodes all the sequence embeddings in parallel with bi-directional attention, leading to more efficient and effective predictions.

## 3.4 MULTI-TASK TRAINING

**Loss Function.** We adopt a softmax cross-entropy loss to maximize the likelihood of masked sequence conditioned on the image feature:

$$L = \sum_t W_t \frac{1}{N_m} \sum_{i \in M} \log P(\hat{y}_i | x, y) \qquad (1)$$

where $y$ and $\hat{y}$ are masked and decoded sequences, $W_t$ is loss weights for different tasks, $M$ means the set of masked tokens, and $N_m$ denotes the number of masked tokens. Only the loss of masked tokens is counted. Following previous practice (Carion et al., 2020; Al-Rfou et al., 2019), we adopt auxiliary losses for the predictions after each decoder layer. For each task, we filter the target token vocabulary so that losses are only calculated on its involved vocabulary to improve training efficiency. Considering tokens of the whole vocabulary leads to intensive computation and memory usage since image captioning involves plenteous text tokens that are not involved in other tasks.

**Task Mixed Sampling.** For learning a single model for multiple tasks, we employ a task mixed sampling strategy where each image in the dataset is sampled with its annotations mixed from all tasks. The sampled images are processed by the backbone and encoder only once for encoding image features shared by all tasks. Only the decoding process is repeated for different tasks, considering that they hold different sequence lengths and are hard to process in parallel. Such a strategy is conceptually simple and effective compared with the batch mixing strategy from existing work (Chen et al., 2022a; Li et al., 2023) where each batch only samples image-sequence pairs for a single task. Considering that each image might involve multiple vision tasks, batch mixing requires encoding the same image multiple times for different tasks. As a comparison, task mixing provides a more flexible framework to add more data from more tasks, while also sharing most model components among tasks, resulting in better efficiency.

## 4 EXPERIMENTS

### 4.1 EXPERIMENTAL SETTINGS

**Dataset and Tasks.** Following previous practive (Chen et al., 2022a; Wang et al., 2023), we evaluate MAD on MS-COCO dataset (Lin et al., 2014) which contains 118k training images and 5k validation images with annotations for all four tasks we considered. For object detection, we take $N = 100$ instances per image for training, resulting in a sequence of length 500. The coordinates of bounding boxes are discretized into 500 bins. For instance segmentation, we randomly sample ten instances and

Table 1: Comparisons for object detection ($AP$), instance segmentation ($AP$), keypoint detection ($AP$), and image captioning ($BLEU@4$ (Papineni et al., 2002)) on COCO validation set.

| | Backbone | Param. | Det. | Seg. | Kpt. | Cap. |
|---|---|---|---|---|---|---|
| *Task-specific Models* | | | | | | |
| Faster R-CNN (Ren et al., 2015) | R101-FPN | 42M | 42.0 | - | - | - |
| DETR (Carion et al., 2020) | R101-DC5 | 60M | 44.9 | - | - | - |
| Pix2Seq (Chen et al., 2021) | R101-DC5 | 57M | 45.0 | - | - | - |
| Mask R-CNN (He et al., 2017) | X101-FPN | 107M | 42.9 | 38.6 | - | - |
| Keypoint R-CNN (Wu et al., 2019) | R50-FPN | 59M | - | - | 65.5 | - |
| Transformer (Sharma et al., 2018) | Encoder | - | - | - | - | 34.0 |
| *Generalist Models* | | | | | | |
| VisionLLM (Wang et al., 2023) | R50+Alpaca-7B | 40M + 7B | 44.6 | 25.1 | - | 31.0 |
| Pix2SeqV2 (Chen et al., 2022a) | VIT-B | 132M | 46.5 | 38.2 | 64.8 | 34.9 |
| **MAD (Ours)** | Swin-B | 107M | 49.7 | 40.6 | 64.6 | 32.2 |

transform their segmentation masks into bit masks with a size of $16 \times 16$. For keypoint detection, we train MAD on ten person instances per image and only predict keypoints for detected humans (based object detection results) during inference. We pad blank instances for the above three tasks if there are not enough instances existing in the image. For image captioning, we adopt sentence piece model (SPM) from T5 (Raffel et al., 2020) for tokenization, and abbreviate its vocabulary based on COCO dataset, resulting in 11421 remaining tokens. We use loss weights of [1.5, 2.7, 0.5, 0.3] for object detection, instance segmentation, keypoint detection, and image captioning respectively.

At inference time, we first predict task sequences for object detection as they will serve as the prompt for the subsequent tasks. The detection sequences are decoded into detection results, represented by five tokens including four coordinate tokens and one class token, while the probability of the class token is adopted as the detection score. For instance segmentation, we directly convert predicted sequences into bit masks based on probabilities of $< Foreground >$ tokens. For keypoint detection, the predicted sequences are dequantized into tuples of keypoint coordinates with probabilities of $< Visible >$ showing their visibility. For image captioning, the sequence is truncated by the first padding token and directly mapped back to text by SPM. We conduct masked inference on keypoint detection and image captioinng tasks with mask ratios of 0.7, and 0.8, 0.6, 0.4 respectively.

**Implementation Details.** We implement MAD with two different backbones, Swin-Base for comparison to state-of-the-art methods, and Resnet-50 for ablations. Both the encoder and decoder in Transformer consist of 6 layers with a main dimension of 256 and 8 attention heads, and the width of FFN is set to 2048. For sequence modeling, we adopt learned positional encodings with a length of 506 to cover all task sequences.

We use the AdamW optimizer with an initial transformer's learning rate of 1e-4 and backbone of 1e-5. The batch size is set to 16. For comparisons with state-of-the-art methods, we train the model with Swin-Base (Liu et al., 2021) backbone for 300 epochs with learning rate drop after 200 epochs. For our ablation experiments with the ResNet-50 backbone, we use a shorter training schedule of 50 epochs. We use the same data augmentation strategy consisting of image flipping, randomly resizing, and cropping for all tasks. The input images are re-scaled so that its shortest side is between 480 and 800 pixels while the longest is at most 1333. During inference, the shortest side of the image will be resized to 800 pixels. The inference speeds of all experiments are the total time for inferring on four tasks, tested on a single A100 with a batch size of one image.

## 4.2 COMPARISON WITH STATE-OF-THE-ART METHODS

Tab. 1 shows comparisons with the state-of-the-art (SOTA). We compare MAD with two types of SOTA models: (1) typical task-specific models which leverage task-specific designs and are trained on a single task; (2) generalist models which employ a shared single architecture to handle multiple vision tasks without task-specific designs such as region proposal network (RPN) or ROI Pooling. Compared with the task-specific models, we can see that MAD can achieve competitive and even better accuracy without customized architecture for a single task. On top of that, the sequence-based framework in MAD provides significant scalability and flexibility to new tasks or data formats than these models. In addition, MAD also outperforms existing generalist models with fewer parameters,

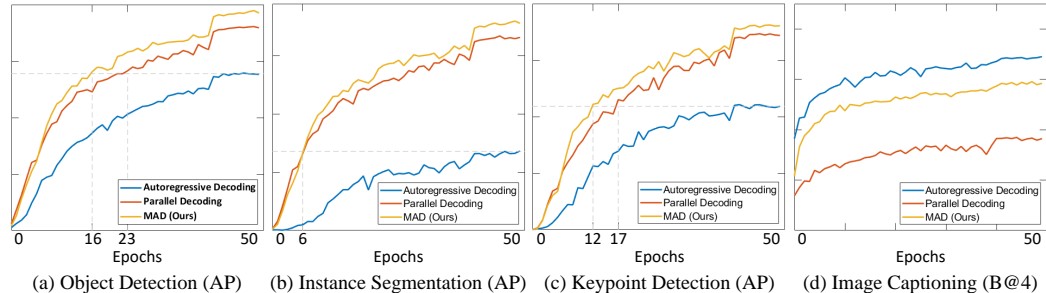

(a) Object Detection (AP)  (b) Instance Segmentation (AP)  (c) Keypoint Detection (AP)  (d) Image Captioning (B@4)

Figure 4: Convergence curves for *Autoregressive Decoding*, *Parallel Decoding*, and the proposed **MAD** in Tab. 2. MAD achieves much faster convergence for vision-centric tasks and greatly narrows the gap with *Autoregressive Decoding* compared with *Parallel Decoding* for image captioning.

Table 2: Ablation studies of MAD. The "(single task)" indicates that the model is separately trained for each single task. The inference time counts the total time of processing all four tasks.

| Methods | Infer. Time (ms) | Det. | Seg. | Kpt. | Cap. |
|---|---|---|---|---|---|
| Autoregressive Decoding | 3953 | 27.9 | 12.3 | 33.4 | 34.1 |
| Parallel Decoding (single task) | - | 38.4 | 31.2 | 55.1 | 20.6 |
| Parallel Decoding | 137 | 35.9 | 29.8 | 51.5 | 18.2 |
| +Masked training | 137 | 38.9 | 32.3 | 54.6 | 18.6 |
| +Masked Inference (**MAD**) | 173 | 38.9 | 32.3 | 54.7 | 29.6 |

especially on vision-centric tasks, demonstrating these tasks greatly benefit from bi-directional attention and masked sequence modeling designs. For image captioning, the autoregressive paradigm in existing methods excels us in modeling language sequential context. We will investigate how to combine the advantages of both in the future for enabling a more versatile generalist model.

## 4.3 ABLATION STUDIES

**Main Components Ablation.** We first gradually ablate our main designs as shown in Tab. 2. We convert MAD into an autoregressive variant with the same architecture for comparison (Details can be found in the supplementary material). It can be seen that *Autoregressive Decoding* performs worst in terms of both inference time and accuracy on vision tasks except image captioning. This result is consistent with our analysis that the autoregressive approach might not fit well for vision-centric tasks and struggles with extremely slow predictions. By employing bi-directional attention and parallel decoding (i.e., *Parallel Decoding*), the convergence and inference speed of vision tasks are greatly improved. However, such a simple parallel decoding method suffers from severe performance degradation compared to its single-task model (i.e., *Parallel Decoding (single task)*), leading to an inferior multi-task learning paradigm.

However, we can observe that introducing our masked sequence modeling during training can significantly mitigate the performance degradation for multi-task learning. As shown in the fourth row, *+Masked training* performs especially better for object detection, instance segmentation, and keypoint detection, thanks to the task context modeled through masking and reconstruction. Moreover, by further introducing masked inference (i.e., *+Masked Inference (MAD)*), the accuracy is constantly improved with competitive image caption accuracy to the autoregressive counterpart. In addition, we observe that MAD has different effects on vision-centric tasks and language tasks in training and inference. We speculate that MAD in training could model rich task contexts, such as the relationship among task prompts, vocabulary, and sequence patterns, which are crucial for modeling multi-task sequences. On the contrary, during inference, MAD mainly focuses on dependencies among sequence tokens, which are generally rich in language but lacking in visual sequences.

**Convergence Curves for Vision Tasks.** Fig. 4 compares the detailed training curves between methods in Tab. 2 for different tasks. With bi-directional attention, both MAD and parallel decoding converge much faster than autoregressive decoding which adopts uni-directional attention. In addition, the

Table 3: Ablation on masked sequence modeling during training. (a) For "random ratio", we used a random mask ratio lying between 0.6 and 0.8. For "multiple ratios", the task sequences are trained with two masking ratios (i.e., 0.6 and 0.8). The "single ratio" indicates that a single mask ratio is adopted. (b) different masking ratios are evaluated under the "single ratio" strategy.

(a) Masking ratio strategies.

| Methods | Det. | Seg. | Kpt. | Cap. |
|---|---|---|---|---|
| random ratio | 38.6 | 31.9 | 54.3 | 29.4 |
| multiple ratios | 38.6 | 32.0 | 54.2 | 29.7 |
| **single ratio** | 38.9 | 32.3 | 54.7 | 29.6 |

(b) Different mask ratios in training.

| Mask Ratio | Det. | Seg. | Kpt. | Cap. |
|---|---|---|---|---|
| 0.4 | 38.4 | 31.8 | 54.4 | 29.2 |
| 0.6 | 38.5 | 31.9 | 54.2 | 29.7 |
| **0.7** | 38.9 | 32.3 | 54.7 | 29.6 |
| 0.8 | 38.7 | 32.1 | 54.9 | 28.4 |

Table 4: Ablations on parameters for individual tasks.

(a) Number of quantization bins for coordinates.

| Number of Bins. | Det. | Kpt. |
|---|---|---|
| 300 | 38.5 | 54.2 |
| **500** | 38.9 | 54.7 |
| 800 | 38.6 | 54.5 |
| 1000 | 38.5 | 54.4 |

(b) Size of bit mask for image segmentation.

| Mask Size | Det. | Seg. |
|---|---|---|
| 12 | 38.4 | 31.5 |
| 14 | 38.8 | 32.0 |
| **16** | 38.9 | 32.3 |
| 20 | 38.8 | 32.4 |

(c) Inference mask ratios for image captioning.

| Mask Ratio | BLUE@4 |
|---|---|
| w/o masked inference | 18.6 |
| {0.7} | 25.8 |
| {0.7, 0.3} | 27.0 |
| **{0.8, 0.6, 0.4}** | 29.6 |

masked sequence modeling strategy in MAD can further capture rich task contexts and largely improve performances, especially for image captioning. These results further demonstrate the non-trivial design of MAD.

**Masked Training.** We examine how varying masking strategies and masking ratios affects the training of MAD. As Tab. 3a shows, the simplest strategy with a single masking ratio could achieve the highest performance. As for specific masking ratios in training (under the *single ratio* strategy), MAD performs the best with a moderate value of 0.7, while a smaller masking ratio results in an over-simplified task, and a larger masking ratio leaves insufficient tokens for modeling task contexts.

**Coordinate Quantization.** We evaluate the effect of the number of the coordinate bins. As Tab. 4a shows, MAD performs robustly under different numbers of bins. We thus adopt 500 as default, while each bin corresponds to approximately 2 pixels for an image with size between 800 to 1333 pixels, resulting in negligible quantization error.

**Mask Size.** In Tab. 4b, we study the size of the segmentation mask. It can be seen that MAD does not benefit much from larger mask sizes, since we do not adopt task-specific operations like ROI Align (He et al., 2017) or interpolation to align mask pixels and image pixels. Considering that larger mask sizes lead to longer task sequences, we set the mask size at 16 for good efficiency.

**Inference Mask Ratio for Captioning.** We examine different inference mask ratios for image captioning. Results in Tab. 4c demonstrate that a combination of gradually decreasing masking ratios ({0.8, 0.6, 0.4}) performs the best.

## 5 CONCLUSION

In this work, we propose Masked AutoDecoder (MAD), a sequence-to-sequence multi-task vision generalist that employs masked sequence modeling and parallel decoding. MAD performs multiple vision tasks with a unified task sequence format, and learns to reconstruct the masked task sequences for modeling diverse task contexts. In addition, we employ bidirectional attention and parallel decoding in Transformer, achieving significant speedup in both convergence and inference compared to autoregressive counterparts for vision tasks. Experiments on COCO demonstrate the effectiveness and superiority of MAD as compared with both well-established task-specific models and existing vision generalist models.

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

# A  APPENDIX

## A.1  AUTOREGRESSIVE DECODING

We convert MAD into an autoregressive variant for comparison. It follows the same architecture as well as training settings as MAD with a few modifications on tokenization of task sequences and decoding process. We adopt same approach of MAD to construct task sequences for instance segmentation, keypoint detection and image captioning, while following pix2seq (Chen et al., 2021) to build detection sequences to match the sequential decoding mechanism of autoregressive approaches. During training, we add $< start >$ tokens before the task sequences and feed them to the decoder, and then supplement the $< end >$ tokens after the task sequence as the target sequence to calculate the loss. The self-attention layer in decoder is modified with uni-directional attention to capture sequential dependencies.

At inference time, the task sequences are recursively generated, starting as the $< start >$ token, and going up to the maximum length corresponding to each task (instead of stopping at the $< end >$ token). We adopt $argmax$ sampling strategy and cache the k-v features in self-attention layer for acceleration. Although some other complex sampling strategies, i.e., beam searching or nucleus sampling (Holtzman et al., 2019) may improve performance, these strategies would further deteriorate currently slow inference speed of autoregressive decoding.

## A.2  TASK WEIGHTING

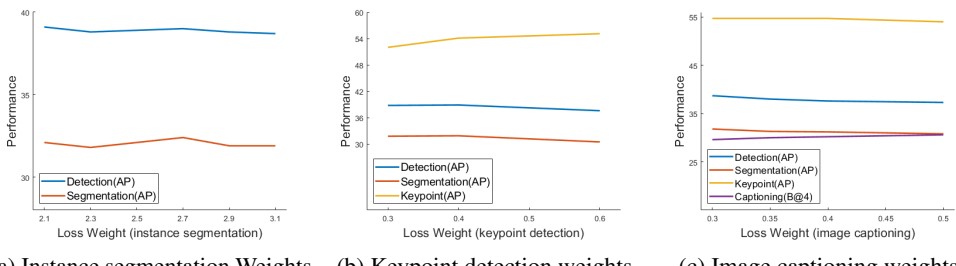

(a) Instance segmentation Weights.    (b) Keypoint detection weights.    (c) Image captioning weights.

Figure 5: Performance with different loss weights by gradually adding new tasks to the existing tasks.

In Fig. 5, we search for the appropriate loss weight for each task. We first evaluate object detection performance and obtain the optimal weight of 1.5, and then introduce the instance segmentation task. As Fig. 5a shows, both tasks perform well over a wide range of weights, with only small fluctuations. We thus simply take a weight of 2.7 for instance segmentation. For keypoint detection, it seems to conflict with the existing tasks, and increasing its weight would hinder other tasks. According to the trade-off of performance, the keypoint detection task is weighted by a factor of 0.5. Finally, we add image captioning task, where we find that a weight of 0.3 to be appropriate for preserving performances of existing vision tasks.

