# OpenReview forum: "Masked AutoDecoder is Effective Multi-Task Vision Generalist"
_ICLR.cc/2024/Conference — ICLR 2024 Conference Withdrawn Submission_

### Official Review · Reviewer_A9Bs · 2023-10-24

**Soundness:** 3 good
**Presentation:** 2 fair
**Contribution:** 2 fair
**Rating:** 5
**Confidence:** 4

**Summary:**

The paper applies the idea of masked autoencoding to the effort of building vision models that can jointly perform multiple tasks (a generalist model). The joint tasks of concern are object detection, segmentation, key-point detection and captioning. And the application of the idea is quite straight-forward at the high-level (Fig 3), that the image will first go through a backbone + encoder pair to get the features from the image as "context". And then a decoder will take the "context" and the masked sequences as inputs to reconstruct the full sequence. During inference, the sequence is predicted iteratively, with the most confident outputs selected as the inputs to the next iteration. The paper outperforms prior works (mainly Pix2Seq) on those tasks with a stronger backbone (Swin-B compared to ViT-B), and some analysis is provided for several important designs (e.g., mask ratio, ways to construct the tokens).

**Strengths:**

+ The direction of building a generalist model for computer vision is interesting, and in my belief a next frontier to explore on the evaluation side following the trajectory of NLP.
+ I believe the idea in the paper works -- meaning that masked autoencoding on the latent feature space should still work, and should help the final performance on multiple tasks.

**Weaknesses:**

- Following the second point of strengths -- unfortunately I think it can be viewed as a weakness of the paper too. Since the common belief is that masked autoencoding should work, it is not presenting new knowledge to me even if it works. I do think the exploration has its value (the value to verify it works), but the value is not too significant, and as time goes by (after people try on many other domains) the value diminishes.
- The writing/presentation is okay but not great. I can follow the gist of the paper, but I think there are certain aspects the paper can definitely be improved. E.g., in Sec 3.1, when talking about task tokenization, the description mingles 1) how the task is tokenized and 2) how the task input is masked. It would be great to revisit that section and make the description decoupled.
- The experimentation is not too convincing to me, especially when doing the system-level comparisons in Table 1. For example, we know that Swin-B is a stronger backbone compared to ViT-B (w.r.t. the tasks of concern here), despite their similarity in number of parameters. Then it is unclear if the improvement compared to Pix2Seq is a result of the efforts paid in this work, or it is simply because Swin-B is just a better backbone.
- I think a lot of the ideas/setups in the paper can be traced back to DETR (or even earlier works), e.g., Hungarian matching, parallel decoding, decoder design (256 dim main, 2048 MLP), I hope the paper could refer to DETR when it was originated there.

**Questions:**

- For the tasks selected (detection, segmentation, keypoint, captioning), Mask R-CNN can already automatically perform all of first three (please check the original Mask R-CNN paper, and the demo from Detectron 1/2). And it is not hard for me to imagine a version for it to perform captioning (or dense captioning, see https://openaccess.thecvf.com/content_cvpr_2016/papers/Johnson_DenseCap_Fully_Convolutional_CVPR_2016_paper.pdf, based on RoI pooling). I wonder what the performance would be like for such a model?

---

### Official Review · Reviewer_qumq · 2023-10-30

**Soundness:** 2 fair
**Presentation:** 3 good
**Contribution:** 2 fair
**Rating:** 3
**Confidence:** 4

**Summary:**

The paper focuses on building a general-purpose transformer for vision and language problems. The authors address this by introducing the Mask AutoDecoder (MAD). They achieve this by introducing a bi-directional attention to capture contextual correlations and a parallel decoding and designing a masked sequence modeling approach that masks and reconstructs task sequences to learn rich task contexts. The experimental results show that the proposed method obtains competitive results compared with previous methods and baselines.

**Strengths:**

* The paper focuses on a practical problem.

* The method is simple and shown to improve the performance.

* The method is shown to be fast.

**Weaknesses:**

* The proposed method is very related and similar to Pix2SeqV2[A]. More specifically, the method can be viewed as introducing a MAD to [A]. Though in Table 1, the proposed method obtains better results than Pix2SeqV2, the proposed method use a different backbone, making it difficult to measure the effectiveness and efficiency of the proposed method. As the authors follow the same setup as the Pix2SeqV2, why not build the proposed method based on Pix2SeqV2? Also, though the ablation study is given, the results of using the proposed MAD for single-task learning are important, which are missing in Table 1.

* In Table 2, by changing the uni-directional attention to the bi-directional one, the multi-task learning method obtains much worse performance than single-task learning while in Pix2SeqV2, the multi-task learning model obtains similar results than single-task learning models. It seems that Pix2SeqV2 performs better for multi-task learning. Also, regarding to Masked inference, which involves multi-stage masked decoding to refine the prediction, one can apply the same to the single-task counterparts and obtains better results.

* The method is very related to [A] which is missing in the paper. I would recommend the authors to discuss the relations and difference compared with [A]

[A] Bachmann et al., MultiMAE: Multi-modal multi-task masked autoencoders, ECCV 2022.

**Questions:**

* Why do the authors use different backbone for comparisons with state-of-the-arts and ablation study?

* Does the proposed method shown in Table 1 use the masked inference during testing?

* What's the image resolution used in this paper? I would recommend the authors to given more details about the experiments.

---

### Official Review · Reviewer_1kN4 · 2023-10-30

**Soundness:** 3 good
**Presentation:** 2 fair
**Contribution:** 2 fair
**Rating:** 5
**Confidence:** 4

**Summary:**

In this paper, going beyond the well-known Pix2Seq V2, the authors further change the multi-step inference in Pix2Seq V2 to the single-step token generation. The results demonstrate that the proposed method is with a relatively good performance compared to Pix2Seq V2. It is a quite interesting discovery that such a small modification can lead to such an enhancement. I believe the insight of building an “efficient general multi-task transformer” beyond Pix2Seq V2 is interesting, but I think the writing failed to describe the method clearly and the experimental results do not support the claim.

**Strengths:**

1) It is quite an interesting discovery that such a small modification can lead to such an enhancement.
3) Easy to understand.

**Weaknesses:**

1) The writing of this paper is not very clear. In the method part, the authors put a lot of effort into introducing the overlap part with Pix2Seq V2, such as the tokenizer and masked modeling. However, the difference from Pix2Seq V2 is not well presented. Given that the ICLR has a quite tight page limit, I am quite astonished that the appendix is very short (I would have thought I could find more details in the appendix). As a result, I am still not very clear on how to achieve the change from multi-step to single-step decoding after reading this paper. This issue would be alleviated by sharing the code or implementation details, but unfortunately not.
2)  I am not sure whether the second contribution (masked sequence modeling) in the abstract is first proposed in this paper. I want to note that the Pix2Seq V2 also proposes a similar approach for training.
3) For the first contribution, the experimental results seem not to support that claim. According to the Table 2, compared to single task decoding, multi-task decoding has relatively lower performance, which is reasonable, but not support the communication between tasks can improve performance. In addition, the efficiency is also not well explored, I did not find any analysis of the computational cost including GFLOPs and FPS in the experiments.

**Questions:**

See Weaknesses

---

### Official Review · Reviewer_iGd5 · 2023-10-31

**Soundness:** 3 good
**Presentation:** 3 good
**Contribution:** 3 good
**Rating:** 6
**Confidence:** 2

**Summary:**

The paper presents Masked AutoDecoder (MAD), a vision model that uses masked sequence modeling and parallel decoding to handle various vision tasks. Unlike existing models that use autoregressive Transformers, MAD introduces bi-directional attention to capture contextual dependencies and decode vision task sequences in parallel. It also uses a masked sequence modeling approach to learn rich task contexts by masking and reconstructing task sequences. As a result, MAD can handle different vision tasks with a single network branch and a simple cross-entropy loss, minimizing the need for task-specific designs. The experiments show that MAD performs better and is more efficient than autoregressive counterparts and achieves competitive accuracy with task-specific models.

**Strengths:**

1 The idea of combining a non-autoregressive method with the pix2seq framework is very original and appropriate.

2 The experimental results are very solid, and the superior performance and inference efficiency of the MAD model over autoregressive models are demonstrated through various vision tasks such as object detection, instance segmentation, keypoint detection, and image captioning.

3 The paper is well-written and the content is well-organized. The concepts and methodology are discussed in detail, making it easy to understand. The use of diagrams further enhances the clarity of the paper.

**Weaknesses:**

1  Lack of Comparison in Inference Efficiency: The paper could benefit from a more comprehensive comparison of the inference efficiency of the proposed MAD model with other methods. While Table 1 provides some comparison, it would be beneficial to see a more detailed analysis, including various models and methods.

2  Limited Ablation Studies on Inference Strategies: The paper could further improve by providing more ablation studies on the inference strategies. Understanding how different aspects of the inference phase impact the overall performance would provide valuable insights into the effectiveness of the proposed model and could potentially lead to further improvements.

**Questions:**

N/A